# Impact of a Rounding Checklist Implementation in the Trauma Intensive Care Unit on Clinical Outcomes

**DOI:** 10.3390/healthcare12090871

**Published:** 2024-04-23

**Authors:** Dongmin Seo, Inhae Heo, Jonghwan Moon, Junsik Kwon, Yo Huh, Byunghee Kang, Seoyoung Song, Sora Kim, Kyoungwon Jung

**Affiliations:** 1Division of Trauma Surgery, Department of Surgery, Ajou University School of Medicine, Suwon 16499, Republic of Korea; suh.d.min@gmail.com (D.S.); soyo1226@naver.com (J.M.);; 2Ajou University Hospital Gyeonggi South Regional Trauma Center, Suwon 16499, Republic of Korea

**Keywords:** severe trauma, outcomes, checklist, mortality, implementation, quality improvement

## Abstract

We aimed to evaluate the effectiveness of an intensive care unit (ICU) round checklist, FAST HUGS BID (Feeding, Analgesia, Sedation, Thromboembolic prophylaxis, Head-of-bed elevation, Ulcer prophylaxis, Glycemic control, Spontaneous breathing trial, Bowel regimen, Indwelling catheter removal, and De-escalation of antibiotics—abbreviated as FD hereafter), in improving clinical outcomes in patients with severe trauma. We included patients admitted to our trauma ICU from 2016 to 2020 and divided them into two groups: before (before-FD, 2016–2017) and after (after-FD, 2019–2020) implementation of the checklist. We compared patient characteristics and clinical outcomes, including ICU and hospital length of stay (LOS) and in-hospital mortality. Survival analysis was performed using Kaplan–Meier curves and multivariable logistic regression models; furthermore, multiple linear regression analysis was used to identify independent factors associated with ICU and hospital LOS. Compared with the before-FD group, the after-FD group had significantly lower in-hospital mortality and complication rates, shorter ICU and hospital LOS, and reduced duration of mechanical ventilation. Moreover, implementation of the checklist was a significant independent factor in reducing ICU and hospital LOS and in-hospital mortality. Implementation of the FD checklist is associated with decreased ICU and hospital LOS and in-hospital mortality.

## 1. Introduction

Patients with severe trauma require intensive care unit (ICU) treatment for various complex injuries. However, the intricate nature of the treatment process in trauma care creates an environment prone to medical errors, which is often referred to as a “perfect storm”. These errors may originate from diverse factors, including fluctuations in vital signs, incomplete documentation of patient medical histories, and inadequate availability of information. In addition, trauma surgeons face the challenge of making time-critical decisions while simultaneously managing complex teams and collaborating with professionals from many disciplines [1]. When managing patients with severe injury, trauma surgeons usually play a dual role as both surgical specialists and critical care providers, and they are responsible for all treatment processes. Therefore, trauma surgeons should possess the necessary qualifications and capabilities in surgical critical care in order to manage their patients in the ICU. Maintaining consistent quality of surgical services and ensuring seamless continuity in patient management can significantly enhance overall outcomes [2]. Furthermore, it is crucial for trauma surgeons to minimize medical errors and provide high-quality critical care while performing resuscitation and surgical procedures in the trauma bay and operating room, respectively.

Numerous studies have demonstrated the significance of appropriate physician staffing as well as the use of protocols, clinical practice guidelines, and checklists to improve clinical outcomes in modern ICUs [3,4,5]. Specifically, some studies have shown that the implementation of checklists for organized critical care in ICUs improved patients’ outcomes [6,7]. In our trauma ICU, we recently adopted the Feeding, Analgesia, Sedation, Thromboembolic prophylaxis, Head-of-bed elevation, Ulcer prophylaxis, Glycemic control, Spontaneous breathing trial, Bowel regimen, Indwelling catheter removal, and De-escalation of antibiotics (FAST HUGS BID) checklist, which can be easily remembered through simple mnemonics. This checklist was initially introduced as “FAST HUG” by Vincent in 2005 and later updated to “FAST HUGS BID” by Vincent III and Hatton in 2009 [8,9]. The FAST HUGS BID (abbreviated as FD hereafter) is a multidisciplinary protocol and checklist that includes essential and evolving components of evidence-based critical care. 

Studies have demonstrated that the implementation of the FD checklist substantially affects patients with trauma in the ICU. Stahl et al. showed that a structured checklist could reduce medical errors in the management of patients with trauma [6], Barcellos et al. reported that its implementation reduced the duration of mechanical ventilation and ICU stay [10], and Pronovost et al. demonstrated that its implementation decreased the rate of catheter-related bloodstream infections [11]. Additionally, some studies have demonstrated a reduction in the rate of ventilator-associated pneumonia (VAP) following the use of a checklist [12,13]. Haynes et al. also reported a decrease in mortality and complications following intraoperative utilization of a checklist [14,15]. 

Accordingly, we hypothesized that the implementation of the FD checklist in the trauma ICU would reduce in-hospital mortality and length of stay (LOS) in the ICU and hospital. To this end, we aimed to compare clinical outcomes before and after the implementation of the FD checklist in our trauma ICU; furthermore, we aimed to assess the effectiveness of each checklist component before and after its implementation. 

## 2. Materials and Methods

### 2.1. Study Procedure

This retrospective observational study included patients admitted to the trauma ICU at Ajou University Hospital from March 2016 to December 2020. We excluded patients who were aged <18 years, transferred from other hospitals, admitted to the general ward, or classified as dead on arrival. Eligible patients were divided into two groups based on different time phases: before implementation of the FD checklist (before-FD; from March 2016 to December 2017) and after implementation (after-FD; from January 2019 to December 2020). The FD checklist was implemented at our trauma center in mid-2018. The FD protocol was modified to align with the specific requirements of our trauma ICU, with these modifications being communicated to all team members. These details have been included as a Appendix A. Although it is challenging to ensure complete adherence to all checklist items, we meticulously trained our team members on the protocol and implemented monitoring mechanisms to ensure compliance. Notably, the trauma ICU at Ajou University Hospital operates as a semi-closed ICU, with dedicated trauma surgeons conducting bedside rounds more than twice daily. Furthermore, the FD checklist was prominently displayed at each patient’s bedside and applied to all ICU patients by the trauma staff, nurses, and other team members. 

### 2.2. Definition and Study Outcomes

The FD is a checklist that highlights the key factors in the general care of critically ill patients. This approach includes the following clinical practices: feeding, analgesia, sedation, thromboembolic prophylaxis, head-of-bed elevation, ulcer prophylaxis, glycemic control, spontaneous breathing trials, bowel regimens, indwelling catheters, and drug de-escalation. 

The primary outcomes were the in-hospital mortality, overall complication rate, ICU LOS, hospital LOS, and duration of invasive mechanical ventilation. Secondary outcomes included the incidence of each complication, such as acute kidney injury, acute respiratory distress syndrome, pneumonia, venous thromboembolism, pressure ulcer, surgical site infection, urinary tract infection, catheter-related bloodstream infections, or sepsis. Additionally, we conducted a detailed analysis by comparing the impact of each component of the checklist before and after the intervention.

### 2.3. Statistical Analysis

Categorical variables are presented as proportions and were compared before and after FD using the chi-squared test. Continuous variables are presented as means with standard deviations or medians with interquartile ranges. Between-group comparisons of continuous variables were conducted using Student’s *t*-test for those with a normal distribution and the Mann–Whitney U test for those with a non-normal distribution. Kaplan–Meier curves and logistic regression models were employed to perform survival analysis. Multiple linear regression analysis was conducted to identify factors independently associated with ICU and hospital LOS. All variables with a *p* value < 0.1 in the univariable analysis were included in the multivariable model. The threshold for statistical significance was set at *p* < 0.05. Statistical analyses were performed using SPSS 25.0 for Windows (SPSS, Inc., Chicago, IL, USA). 

## 3. Results

### 3.1. Characteristics of Patients

We reviewed 12,959 patients admitted to the trauma ICU during the study period; among them, 10,543 patients were excluded based on the criteria. Among the 2416 patients included in the final sample, 696 and 1720 were included in the before-FD and after-FD groups, respectively (Figure 1). Patient characteristics are summarized in Table 1. The after-FD group exhibited a higher prevalence of underlying diseases and a greater Injury Severity Score (ISS) than the before-FD group (*p* < 0.05). 

### 3.2. Primary Outcomes

Compared with the before-FD group, the after-FD group had lower in-hospital mortality (8.3% vs. 4.8%; *p* < 0.05) and complication rates (23.0% vs. 16.5%; *p* < 0.05). In addition, the after-FD group exhibited shorter ICU LOS (7.8 days vs. 5.1 days; *p* < 0.05), hospital LOS (24.3 days vs. 17.6 days; *p* < 0.05), and duration of invasive mechanical ventilation (9.2 days vs. 5.0 days; *p* < 0.05) (Table 2) than the before-FD group. 

### 3.3. Secondary Outcomes

The after-FD group had lower complication rates for pressure ulcers (10.9% vs. 5.7%; *p* < 0.05), pneumonia (9.1% vs. 3.4%; *p* < 0.05), and surgical site infections (4.3% vs. 1.3%; *p* < 0.05) than the before-FD group. However, a higher incidence of sepsis was observed in the after-FD group than in the before-FD group (0.4% vs. 1.5%; *p* < 0.05) (Table 3). 

### 3.4. Comparisons of Each Component of the FAST HUGS BID Checklist 

Each component of the checklist was compared between the before-FD and after-FD groups (Table 4). For feeding, the after-FD group exhibited a faster time to the first start of enteral nutrition and greater body weight gain than the before-FD group (*p* < 0.05). For analgesia, the after-FD group had a shorter duration of intravenous (IV) fentanyl use, higher usage of IV nefopam, and increased use of per oral analgesics compared with the before-FD group (*p* < 0.05). For sedation, the after-FD group showed a lower rate of IV midazolam, IV propofol, and IV vecuronium use, as well as a higher rate of IV dexmedetomidine use, than the before-FD group (*p* < 0.05). For thromboembolic prophylaxis, the after-FD group had a higher rate of SC enoxaparin use and a shorter time to the first use of enoxaparin than the before-FD group (*p* < 0.05). For head-of-bed elevation, the after-FD group had a shorter time to the first start of head-of-bed elevation than the before-FD group (*p* < 0.05). For ulcer prophylaxis, the after-FD group exhibited a lower rate of ulcer medication and pantoprazole use than the before-FD group (*p* < 0.05). For spontaneous breathing trial, the after-FD group showed a shorter time to extubation and a higher rate of unplanned intubation events than the before-FD group (*p* < 0.05). For bowel regimen, the after-FD group demonstrated a lower rate of diarrhea than the before-FD group (*p* < 0.05). For indwelling catheter removal, the after-FD group showed a shorter time to central venous catheter removal and urinary catheter removal than the before-FD group (*p* < 0.05). For drug de-escalation, the after-FD group had a lower rate of restricted antimicrobial use and a shorter duration of antimicrobial use than the before-FD group (*p* < 0.05). 

### 3.5. Factors Associated with In-Hospital Mortality and LOS in the ICU and Hospital

Multivariable logistic regression analysis showed that the implementation of the FD checklist was an independent factor associated with in-hospital mortality (adjusted OR = 0.434; *p* = 0.008). Other factors associated with in-hospital mortality included the initial Glasgow Coma Scale (GCS), ISS, and complications (Table 5). The Kaplan–Meier curve for the comparison of the 90-day in-hospital mortality indicated that the after-FD group had a significantly higher survival rate than the before-FD group (*p* = 0.002) (Figure 2). 

In addition, multivariate linear regression analysis revealed that the implementation of the FD checklist was an independent contributing factor for LOS in the ICU (B = −0.118; *p* < 0.001) and hospital (B = −0.063; *p* = 0.002). Moreover, the initial GCS score, ISS, and complications were found to be associated with ICU and hospital LOS (Table 6 and Table 7).

## 4. Discussion

Our findings indicated that the implementation of the FD checklist improved the clinical outcomes of patients with trauma in the ICU. Although patients in the after-FD group had relatively higher severity of injuries, as indicated by a greater ISS, they exhibited lower rates of in-hospital mortality and complications, shorter ICU and hospital LOS, and reduced duration of invasive mechanical ventilation compared with those in the before-FD group. Furthermore, multivariate logistic and linear regression analyses revealed that the implementation of the FD checklist was an independent factor associated with shorter LOS in the trauma ICU and hospital as well as lower in-hospital mortality.

There are several factors demonstrating that the implementation of the FD checklist substantially affects the trauma ICU population. First, the implementation of the checklist resulted in a decrease in medical errors due to its concise and easily memorable format [6], which included the fundamental components of care provided by all members of the trauma team. Second, the trauma team could make daily care plans and assess their completion using the checklist during daily ICU rounds [10,11,12,13,14,15]. Several studies have endorsed our approach; however, there have been scarce comparative studies evaluating the impact of the FAST HUG or its updated version since its initial proposal by Vincent [8,9]. Additionally, these studies primarily focused on comparing outcomes of specific diseases or individual components, including VAP reduction. In contrast, we selected indicators that could assess the effect of each element within the FD, collected data, and compared their applications. In addition, we specifically compared LOS in the ICU and hospital as well as in-hospital mortality using the Kaplan–Meier curve, which are considered quality indicators for critical care treatment. Furthermore, we employed multivariate logistic regression analysis to demonstrate the association between the application of the FD and other relevant factors. Our findings indicate that implementation of the FD, along with other factors, has a significant impact on clinical outcomes. 

*F for Feeding*. Malnutrition or weight loss increases complications and worsens outcomes in severely ill patients [16]. Li et al. found that early enteral feeding in patients with trauma in the ICU was associated with lower mortality and shorter hospital LOS [10]. Additionally, Ortiz-Reyes et al. suggested that compared with delayed enteral nutrition, early enteral nutrition improved clinical outcomes in mechanically ventilated patients [17]. Moreover, early enteral nutrition has been associated with lower mortality, shorter LOS, and improved clinical outcomes in mechanically ventilated patients, especially among patients with trauma [18]. These findings are consistent with our findings, where the after-FD group exhibited a shorter time to initiate enteral nutrition than the before-FD group. Notably, feeding represents one of the most important advantages of the FD since patients with sepsis or trauma may require nearly double the amount of energy during the acute phase [19]. 

*A for Analgesia*. Pain can affect a patient’s psychological and physiological recovery, and adequate pain relief is an integral part of effective intensive care management. Critically ill patients commonly experience pain due to not only their underlying illness but also routine procedures, such as turning, suctioning, and dressing changes [20]. After implementing the FD checklist at our center, we observed a decrease in the use of narcotic analgesics and an increase in the utilization of nefopam. Despite maintaining a pain score of three points in both groups, pain was effectively managed without the excessive use of narcotic analgesics, which has been considered a positive effect [21,22,23]. Accordingly, it can be argued that the elimination of disadvantages associated with the misuse and abuse of narcotic analgesics should be prioritized over concerns about the potential side effects arising from the increased use of nefopam.

*S for Sedation*. Although the after-FD group exhibited a higher rate of dexmedetomidine and propofol usage as well as a lower rate of benzodiazepine usage, there were similar sedation levels in both groups, which was demonstrated by Richmond Agitation-Sedation Scale scores consistently falling within the target range throughout this study. Previous studies have demonstrated that dexmedetomidine or propofol offers advantages (e.g., reduced ventilation days and LOS in the ICU) over benzodiazepines in sedating critically ill patients [24,25]. It may be difficult and controversial to conclude that the use of dexmedetomidine or propofol instead of benzodiazepines directly affects survival in critically ill patients [24]. However, based on the findings of our study, it can be inferred that reducing benzodiazepine usage while increasing dexmedetomidine or propofol usage contributed to a decrease in the duration of mechanical ventilation, ICU LOS, and overall mortality. Notably, a recent meta-analysis by Ng et al. reported the effectiveness of dexmedetomidine in reducing the incidence of delirium or agitation in the ICU [26]. However, given the retrospective nature of this study, there was a limitation in the between-group comparison of the occurrence of delirium.

*T for Thromboembolic Prophylaxis*. According to the recently published American Association for the Surgery of Trauma and the American College of Surgeons Committee on Trauma clinical guidelines, patients with severe trauma face a high risk of venous thromboembolism, and prioritizing prophylaxis is crucial in preventing potentially lethal complications [27]. Indeed, appropriate thromboembolic prophylaxis for critically ill patients with severe trauma who are unable to move or walk is an important factor in achieving favorable outcomes [27]. In this study, we confirmed that thromboembolic prophylaxis was more actively administered after the implementation of the FD approach; moreover, there was a higher utilization rate of low molecular weight heparin and enoxaparin as well as a shorter time to the initial administration of enoxaparin in the after-FD group. 

*H for Head-of-Bed Elevation*. In 1992, Torres et al. demonstrated that elevating the head-of-bed for patients reduces the incidence of gastroesophageal reflux and VAP in ventilated patients [28]. A recent systematic review indicated that head-of-bed elevation to 30°–60° resulted in a reduced occurrence of VAP (with an absolute risk reduction of 25.7%) compared with supine (0°–10°) positioning. However, the analysis did not reveal any improvement in other outcomes, including microbiologically proven VAP, length of ICU and hospital stay, or duration of mechanical ventilation [29]. There are various opinions regarding the optimal degree of head-of-bed elevation (30°–60°), but it is unclear which degree is most effective in reducing VAP [30]. After the implementation of the FD checklist, we uniformly applied an elevation of 30° in all patients, except for those with cerebrospinal fluid leakage due to unstable spinal cord injury or skull base fracture before fixation. We maintained the angle for head-of-bed elevation at 30° in the trauma ICU since we observed an increase in the incidence of unplanned tube removal (e.g., endotracheal tube, central venous catheter, and chest tube) when the elevation exceeded 30°. Furthermore, we witnessed an increase in pressure sores resulting from back maceration, which were primarily caused by the patient sliding down the bed. Moreover, we confirmed that the after-FD group had a shorter time to initiate head-of-bed elevation and a significantly reduced incidence of pneumonia. Although other elements were also affected, it can be inferred that the uniform and systematic implementation of head-of-bed elevation contributed to the reduced incidence of VAP. 

*U for Ulcer Prevention*. Before 2020, routine prophylaxis against stress ulcers in the ICU was not well justified; furthermore, the advantages and disadvantages of stress ulcer prevention remained unclear [31,32]. However, a recent review published in 2022 concluded that prophylaxis should be considered for critically ill patients with risk factors for stress ulcers. Nonetheless, frequent reassessment and de-escalation of therapy are required when patients have a decreased risk of bleeding [33]. In our study, the after-FD group showed a lower rate of utilization of ulcer medication, including proton pump inhibitors, than the before-FD group. This could be attributed to the selective use of medications and the de-escalation of stress ulcer prevention therapy through regular re-evaluation in the trauma ICU following the implementation of the FD checklist.

*G for Glucose Control*. A recent systematic review highlighted that intensive glucose control is related to an increased risk of severe hypoglycemic events; however, it also leads to reduced ICU LOS, sepsis, and mortality [34]. In this study, blood glucose levels were within the optimal range in both the before-FD and after-FD groups. However, the average blood sugar level was lower in the after-FD group than in the before-FD group, suggesting that blood sugar was controlled more efficiently in the after-FD group. Although this study had limitations in directly confirming whether better-controlled blood glucose levels were associated with reduced ICU and hospital LOS, sepsis incidence, or mortality, it can be inferred that these factors might have indirectly influenced clinical outcomes.

*S for Spontaneous Breathing Trial*. A randomized controlled trial published in 2008 demonstrated a significant reduction in ventilation days, ICU and hospital LOS, and mortality when spontaneous awakening and breathing trials were implemented daily [35]. Additionally, Robertson et al. showed that the implementation of a daily spontaneous breathing trial protocol improved extubation rates [36]. In this study, we observed that the after-FD group had a shorter time to extubation but a higher rate of unplanned intubation than the before-FD group. Herein, we need to interpret with caution that the before-FD group exhibited an unusually low rate (0.9%) of unplanned intubation, which is exceptionally lower than the optimal range for failed extubation (5–10%) [37]. These findings might also be related to more cautious practices regarding extubation attempts. 

*B for Bowel Regimen*. Diarrhea is common in ICU and is associated with increased mortality rates and prolonged ICU and hospital LOS [38]. In this study, we observed a decrease in the incidence of diarrhea events in the after-FD group. It can be inferred that the ICU staff checked for the presence and frequency of diarrhea daily, changed the feeding solution and diet, or prescribed symptom-relieving medications more aggressively following the checklist’s implementation. These efforts, either directly or indirectly, might have contributed to the decrease in ICU and hospital LOS, as well as the mortality rate, in the after-FD group. 

*I for Indwelling Catheter Removal*. Central venous and urinary catheters are necessary for massive transfusion, drug administration, total parenteral nutrition, and close monitoring of critically ill patients in the ICU. However, the risk of infection associated with these catheters is a critical concern. Therefore, numerous studies, including recent systematic reviews, have highlighted the importance of promptly removing unnecessary central venous and urinary catheters in order to prevent catheter-associated infections [39,40,41,42]. In our study, there were significantly decreased indwelling periods of the central and urinary catheters in the after-FD group. This observation indicates that the condition of the catheter was checked more closely, and unnecessary catheters were removed earlier in each round after implementation of the checklist. Consequently, these efforts might have reduced the LOS in the ICU and hospital as well as reduced mortality.

*D for Drug De-escalation of Antibiotics*. Empirical antibiotic therapy is widely employed in the ICU due to the high prevalence of infections, and appropriate antimicrobial therapy is crucial for critically ill patients. However, caution must be exercised since it may cause the development of drug-resistant organisms. Consequently, Masterton emphasized the importance of prompt antibiotic de-escalation and discontinuation [43]. A recent systematic review demonstrated that antibiotic de-escalation therapy can be both safe and effective for most infections [44]. Consistent with these previous findings, our findings indicated a significant decrease in the number of antibiotics used, especially restricted antimicrobials typically reserved for drug-resistant organisms, and a shorter duration of antibiotic use in the after-FD group. These results suggest that antibiotic utilization was managed more closely and effectively after implementation of the checklist.

To our knowledge, this is the first study to evaluate the effectiveness of the implementation of the ICU round checklist, FD, in patients with severe trauma. Notably, our study differs from previous studies in that we analyzed each checklist component separately to identify potential improvements following its implementation. In addition, this study has several strengths, including the quantitative assessment of how much the checklist contributed to reducing ICU and hospital LOS as well as improving in-hospital mortality by constructing a multivariable logistic regression model with adjustment for various confounding factors. 

This study has limitations given its retrospective observational nature. Regarding the evaluation of performance and outcome improvement, various factors other than the checklist’s introduction might have influenced the results. For better comparison and outcome derivation, it may be necessary to perform subgrouping based on certain key factors, followed by additional analysis. For example, patients with gastrointestinal or head injury may have similar ISS but different timings of feeding. Additionally, the need for sedation and analgesics differed between those who underwent surgery or conservative treatment despite both groups possibly having a similar ISS. However, there was controversy regarding the criteria for dividing subgroups; moreover, some subgroups had an insufficient sample size, which impeded comparative analyses. Furthermore, we could not find out the reason why a higher incidence of sepsis was observed in the after-FD than in the before-FD. This finding may be attributed to the fact that patients after FD had a higher injury severity and a higher prevalence of underlying disease, but the exact reason is still unclear due to the limitations of a retrospective analysis. Additionally, the presence of a time gap between the before- and after-FD periods might have impeded accurate comparison. Partial implementation of the checklist began in early 2018, with full implementation being established by 2019. This interim phase was considered as part of the checklist’s implementation process; accordingly, we excluded data from 2018 in order to prevent bias in the before–after comparison since the FD checklist was being refined through trial and error during that period. Although it is difficult to ensure complete adherence to all checklist items, we meticulously trained our team members on the protocol and implemented monitoring mechanisms to ensure compliance. However, we concede that our data may not fully capture the extent of checklist adherence. The difference in group sizes might have increased the possibility of selection bias. The increased sample size in the after-FD period could be attributed to the growing maturity of the trauma center, resulting in an influx of severely injured patients with trauma. Moreover, since this study was conducted at a single center exclusively for patients with trauma, the generalizability of our results to other ICU settings may be limited. 

## 5. Conclusions

The ICU round checklist, FAST HUGS BID, has been shown to be a valuable tool in enhancing communication and minimizing the risk of vital information being overlooked during discussions regarding the care of patients with severe trauma. In this study, we confirmed improvements in each component of the checklist following its introduction in the trauma ICU. Moreover, we observed that its implementation was associated with a decrease in in-hospital mortality as well as ICU and hospital LOS. These findings highlight the potential of utilizing a round checklist in an intensive care setting for severely injured patients, as it effectively reduces risks and improves clinical outcomes.

## Figures and Tables

**Figure 1 healthcare-12-00871-f001:**
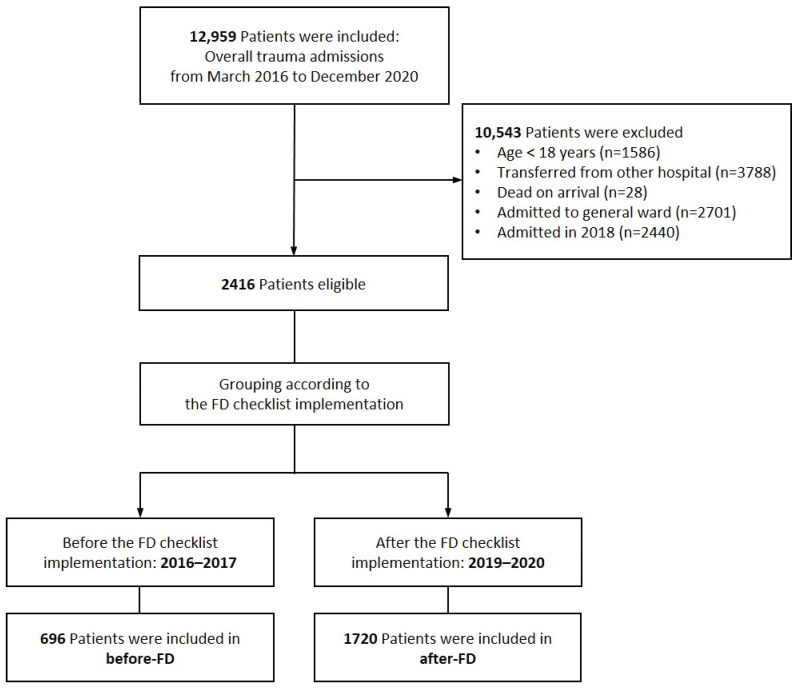
Flow chart of patient selection. Before-FD, before implementation of the FAST HUGS BID checklist; after-FD, after the implementation of the FAST HUGS BID checklist.

**Figure 2 healthcare-12-00871-f002:**
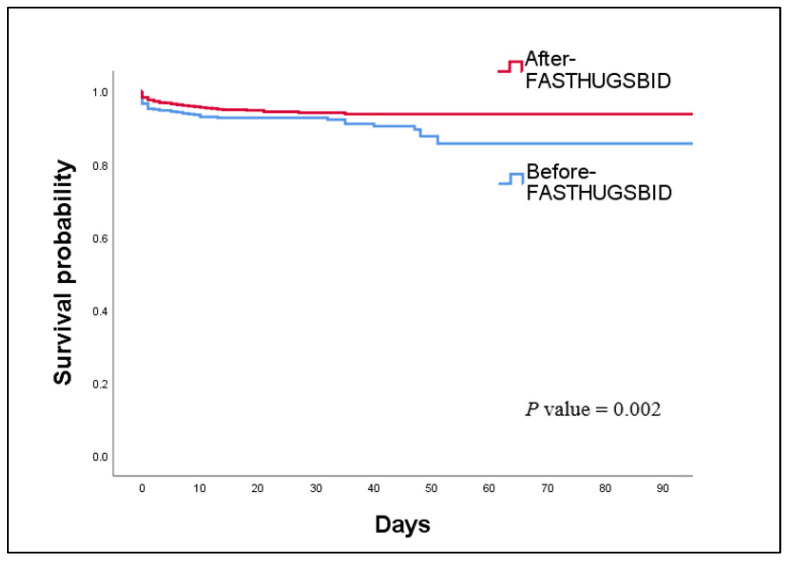
Kaplan–Meier curve for comparison of 90-day in-hospital mortality before and after implementation of the FAST HUGS BID program.

**Table 1 healthcare-12-00871-t001:** Patient characteristics in the before- and after-FD groups.

Variables	Before-FD ^1^ (*n* = 696)	After-FD ^2^ (*n* = 1720)	*p* Values
Age (year), mean ± SD ^3^	48.7 ± 17.5	48.7 ± 16.8	0.97
Sex, *n* (%)			0.09
Female	171 (24.6)	368 (21.4)	
Male	525 (75.4)	1352 (78.6)	
Mechanism of injury			0.539
Blunt, *n* (%)	618 (89.3)	1545 (90.1)	
Penetrating, *n* (%)	74 (10.7)	169 (9.9)	
Underlying disease, yes (%)	263 (38.3)	734 (42.9)	<0.05
Injury Severity Score, median (IQR ^4^)	13 (5–22)	17 (10–24)	<0.05
Initial physiologic parameters			
Systolic blood pressure (mm Hg), mean ± SD	131.2 ± 26.9	136.0 ± 26.7	<0.05
Diastolic blood pressure (mm Hg), mean ± SD	82.2 ± 20.9	89.2 ± 21.1	<0.05
Mean arterial pressure (mm Hg), mean ± SD	98.5 ± 21.4	104.8 ± 21.2	<0.05
Pulse rate (per min), mean ± SD	90.7 ± 19.9	89.2 ± 20.4	0.09
Respiratory rate (per min), mean ± SD	20.4 ± 5.8	21.1 ± 5.9	<0.05
Body temperature (°C), mean ± SD	36.4 ± 0.7	36.5 ± 0.7	0.25
Glasgow Coma Scale, median (IQR)	15 (13–15)	15 (14–15)	0.07

^1^ Before-FD, before implementation of the FAST HUGS BID checklist; ^2^ After-FD, after the implementation of the FAST HUGS BID checklist; ^3^ SD, standard deviation; ^4^ IQR, interquartile range; FAST HUGS BID, Feeding, Analgesia, Sedation, Thromboembolic prophylaxis, Head-of-bed elevation, Ulcer prophylaxis, Glycemic control, Spontaneous breathing trial, Bowel regimen, Indwelling catheter removal, and De-escalation of antibiotics.

**Table 2 healthcare-12-00871-t002:** Comparisons of primary outcomes between the before- and after-FD groups.

Primary Outcomes	Before-FD (*n* = 696)	After-FD (*n* = 1720)	*p* Values
In-hospital mortality, *n* (%)	58 (8.3)	83 (4.8)	<0.05
Complications, *n* (%)	160 (23.0)	283 (16.5)	<0.05
ICU length of stay (days), mean ± SD	7.8 ± 13.3	5.1 ± 10.4	<0.05
Hospital length of stay (days), mean ± SD	24.3 ± 24.6	17.6 ± 16.0	<0.05
Duration of mechanical ventilation (days), mean ± SD	9.2 ± 13.3 (*n* = 315)	5.0 ± 8.4 (*n* = 682)	<0.05

**Table 3 healthcare-12-00871-t003:** Comparisons of secondary outcomes between the before- and after-FD groups.

Complications	Before-FD (*n* = 696)	After-FD (*n* = 1720)	*p* Values	OR ^1^ (95% CI ^2^)
Total, *n* (%)	160 (23)	283 (16.5)	<0.05	0.66 (0.53–0.82)
Acute kidney injury, *n* (%)	9 (1.3)	26 (1.5)	0.68	1.17 (0.54–2.51)
Acute respiratory distress syndrome, *n* (%)	4 (0.6)	9 (0.5)	1.00	0.91 (0.28–2.97)
Pressure ulcer, *n* (%)	76 (10.9)	98 (5.7)	<0.05	0.49 (0.36–0.67)
Venous thromboembolism, *n* (%)	8 (1.1)	24 (1.4)	0.63	1.21 (0.54–2.72)
Pneumonia, *n* (%)	63 (9.1)	59 (3.4)	<0.05	0.38 (0.25–0.52)
Surgical site infection, *n* (%)	30 (4.3)	23 (1.3)	<0.05	0.30 (0.17–0.52)
Urinary tract infection, *n* (%)	8 (1.1)	18 (1.0)	0.82	0.91 (0.39–2.10)
Catheter-related blood stream infection, *n* (%)	2 (0.3)	9 (0.5)	0.74	1.83 (0.39–8.47)
Sepsis, *n* (%)	3 (0.4)	25 (1.5)	<0.05	3.40 (1.02–11.32)

^1^ OR, odds ratio; ^2^ CI, confidence interval.

**Table 4 healthcare-12-00871-t004:** Comparisons of each component of the FAST HUGS BID checklist before and after implementation.

Component of FAST HUGS BID Checklist	Before-FD (*n* = 696)	After-FD (*n* = 1720)	*p* Values	OR (95% CI)
Feeding				
Time to enteral nutrition (days), mean ± SD	2.7 ± 3.7	1.5 ± 1.9	<0.05	-
Time to parenteral nutrition (days), mean ± SD	7.9 ± 8.9 (*n* = 161)	9.2 ± 9.5 (*n* = 209)	0.165	-
Body weight difference (kg), mean ± SD	0.46 ± 6.3	0.52 ± 3.9	<0.05	-
Analgesia				
IV ^1^ fentanyl use, *n* (%)	508 (73.0)	1225 (71.2)	0.382	0.91 (0.75–1.11)
Duration of IV fentanyl use (days), mean ± SD	5.8 ± 7.6 (*n* = 508)	2.5 ± 2.9 (*n* = 1225)	<0.05	–
IV nefopam use, *n* (%)	421 (60.5)	1407 (81.8)	<0.05	2.94 (2.41–3.57)
PO ^2^ painkillers use, *n* (%)	456 (65.5)	1382 (80.3)	<0.05	2.15 (1.77–2.61)
PO opioids use, *n* (%)	72 (10.3)	654 (38.0)	<0.05	5.31 (4.09–6.91)
TD ^3^ fentanyl patch use, *n* (%)	196 (28.2)	550 (32.0)	0.66	1.20 (0.99–1.46)
Pain scale, median (IQR)	1 (0.1–2.0)	1.2 (0.5–2.1)	<0.05	–
Sedation				
IV midazolam use, *n* (%)	201 (28.9)	113 (6.6)	<0.05	0.17 (0.13–0.22)
IV propofol use, *n* (%)	205 (29.5)	434 (25.2)	<0.05	0.80 (0.66–0.98)
IV dexmedetomidine use, *n* (%)	138 (19.8)	452 (26.3)	<0.05	1.44 (1.16–1.79)
IV vecuronium use, *n* (%)	69 (9.9)	83 (4.8)	<0.05	0.46 (0.33–0.64)
RASS ^4^ score, median (IQR)	–0.4 (–1.5–0.0)	–0.3 (–1.0–0.0)	<0.05	–
Thromboembolic prophylaxis				
SC ^5^ LMWH use, *n* (%)	165 (23.7)	690 (40.1)	<0.05	2.16 (1.77–2.63)
Time to the first use of LMWH ^6^ (days), mean ± SD	4.8 ± 4.8 (*n* = 165)	2.8 ± 2.8 (*n* = 690)	<0.05	–
Head-of-bed elevation				
Time to first head-of-bed elevation (days), mean ± SD	8.8 ± 16.8	4.8 ± 12.0	<0.05	–
Ulcer prophylaxis				
H2-blocker use, *n* (%)	640 (92.0)	1476 (85.8)	<0.05	0.53 (0.39–0.72)
Proton pump inhibitor use, *n* (%)	64 (9.2)	86 (5.0)	<0.05	0.52 (0.37–0.72)
Glycemic control				
Average level of blood sugar (mg/dL), mean ± SD	137.1 ± 32.6	134.5 ± 31.3	0.07	–
Hypoglycemic event, *n* (%)	29 (4.2)	67 (3.9)	0.76	0.93 (0.59–1.46)
Spontaneous breathing trial				
Time to extubation (days), mean ± SD	6.4 ± 7.2 (*n* = 308)	4.8 ± 5.5 (*n* = 594)	<0.05	–
Unplanned intubation, *n* (%)	6 (0.9)	37 (2.2)	<0.05	2.53 (1.06–6.02)
Bowel movement				
Diarrhea event, *n* (%)	142 (20.4)	277 (16.1)	<0.05	0.75 (0.60–0.94)
Vomiting event, *n* (%)	108 (15.5)	242 (14.1)	0.36	0.89 (0.70–1.14)
Indwelling catheter				
Time to removal of CVC ^7^ (days), mean ± SD	11.9 ± 16.5	7.6 ± 11.0	<0.05	–
Time to removal of urinary catheter (days), mean ± SD	8.5 ± 16.3	5.8 ± 12.5	<0.05	–
Drug de-escalation				
Restricted antimicrobial use, *n* (%)	175 (25.1)	333 (19.4)	<0.05	0.71 (0.58–0.88)
Duration of antimicrobial use (days), mean ± SD	18.0 ± 22.1	8.2 ± 13.6	<0.05	–

^1^ IV, intravenous; ^2^ PO, per oral; ^3^ TD, transdermal; ^4^ RASS, Richmond Agitation and Sedation Scale; ^5^ SC, subcutaneous; ^6^ LMWH, low-molecular-weight heparin; ^7^ CVC, central venous catheter.

**Table 5 healthcare-12-00871-t005:** Factors associated with in-hospital mortality identified by logistic regression analysis.

Variables	Univariable Analysis		Multivariable Analysis	
	Adjusted OR	*p* Values	Adjusted OR	*p* Values
Age	1.018	<0.001	1.040	<0.001
Sex	0.864	0.499		
Mechanism of injury	0.601	0.148		
Underlying disease	0.933	0.713		
Initial systolic blood pressure	0.985	<0.001	0.898	0.538
Initial diastolic blood pressure	0.976	<0.001	0.764	0.441
Initial mean arterial pressure	0.977	<0.001	1.445	0.482
Initial pulse rate	1.025	<0.001	1.010	0.103
Initial respiratory rate	1.019	0.244		
Initial body temperature	0.458	<0.001	1.048	0.789
Initial Glasgow Coma Scale	0.673	<0.001	0.748	<0.001
Injury Severity Score	1.088	<0.001	1.066	<0.001
FAST HUGS BID	0.551	0.001	0.434	0.008
Complications	4.215	<0.001	2.080	0.016

Nagelkerke $R^{2}\;$ = 0.450, Hosmer–Lemeshow Chi-square test = 9.753, *df =* 8, *p* value = 0.283.

**Table 6 healthcare-12-00871-t006:** Multiple linear regression for factors associated with ICU length of stay.

Variables	Β ^1^	95% CI	Βeta ^2^	*p* Value
Mechanism of injury	0.502	−0.840~1.843	0.014	0.464
Underlying disease	0.897	0.102–1.691	0.042	0.027
Initial systolic blood pressure	−0.409	−0.887~0.069	−1.018	0.094
Initial diastolic blood pressure	−0.791	−1.747~0.164	−1.566	0.104
Initial mean arterial pressure	1.203	−0.229~2.636	2.385	0.100
Initial pulse rate	0.015	−0.008~0.037	0.027	0.202
Initial respiratory rate	0.003	−0.069~0.076	0.002	0.934
Initial body temperature	−0.335	−0.923~0.254	−0.022	0.265
Initial Glasgow Coma Scale	−0.682	−0.841~−0.522	−0.171	<0.001
Injury Severity Score	0.194	0.150~0.238	0.196	<0.001
FAST HUGS BID	−2.742	−3.645~−1.839	−0.118	<0.001
Complications	8.212	7.049~9.374	0.285	<0.001
*R* = 0.500, $R^{2}\;$ = 0.25, adj.$R^{2}\;$ = 0.246				

^1^ Β, unstandardized coefficients; ^2^ Beta, standardized coefficients.

**Table 7 healthcare-12-00871-t007:** Multiple linear regression for factors associated with hospital length of stay.

Variables	β	95% CI	βeta	*p* Value
Initial systolic blood pressure	−40.431	−124.612~43.749	−0.597	0.346
Initial diastolic blood pressure	−97.933	−266.144~70.278	−1.144	0.254
Initial mean arterial pressure	135.708	−116.528~387.944	1.597	0.292
Initial pulse rate	−0.534	−4.176~3.108	−0.006	0.774
Initial body temperature	58.948	−42.239~160.135	0.023	0.253
Initial Glasgow Coma Scale	−189.064	−214.394~−163.734	−0.309	<0.001
Injury Severity Score	28.164	20.962~35.367	0.175	<0.001
FAST HUGS BID	−251.928	−409.369~−94.487	−0.063	0.002
Complications	398.987	200.615~597.359	0.083	<0.001
*R* = 0.430, $R^{2}\;$ = 0.185, adj.$R^{2}\;$ = 0.182				

## Data Availability

The data and materials that support the study findings are available upon reasonable request from the corresponding author (Kyoungwon Jung).

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
