# Peer review of "Impact of a Rounding Checklist Implementation in the Trauma Intensive Care Unit on Clinical Outcomes"

_healthcare, 2024, doi:10.3390/healthcare12090871_

Round 1
Reviewer 1 Report
Comments and Suggestions for Authors
Although implementation of the FAST HUGS BID check list is quite a logical way to improve the clinical outcome of trauma ICU patients, some issues should be addressed before this article can be accepted for publication.
1. What are the protocols for FAST HUGS BID? For example, what is the timing and how to evaluate whether a patient could be subject to early feeding? How to evaluate the adequacy of analgesia and sedation? When to start thromboembolic prophylaxis? What was the target of blood sugar level control? The authors should briefly describe their protocol or at least provide their protocol as a supplement material.
2. The patient groups in Table 1 should be divided into subgroups for better comparison. It is too simplified by just grouping patients by their injury mechanism as blunt/penetrating or just compare the two group of patients with their ISS score. For example, patients with gastrointestinal injury or head injury may have same ISS, but the timing of feeding can be totally different. The need of sedation and analgesics will also be different between those who underwent surgery or conservative treatment even though both group of patients may have a similar ISS.
3. Therefore, patient groups from before-FD and after-FD should be well-matched before undergoing any statistical analysis.
Author Response
Response to Reviewer 1
Response: We sincerely appreciate the reviewers’ constructive comments and suggestions. After careful consideration of the comments, we have endeavored to revise the manuscript accordingly. The point-by-point responses are provided below. The revised sections of the manuscript are highlighted in yellow.
Although implementation of the FAST HUGS BID check list is quite a logical way to improve the clinical outcome of trauma ICU patients, some issues should be addressed before this article can be accepted for publication.
- What are the protocols for FAST HUGS BID? For example, what is the timing and how to evaluate whether a patient could be subject to early feeding? How to evaluate the adequacy of analgesia and sedation? When to start thromboembolic prophylaxis? What was the target of blood sugar level control? The authors should briefly describe their protocol or at least provide their protocol as a supplement material.
Response: Thank you for this comment. Indeed, we modified the FAST HUGS BID protocol to align with the specific requirements of our trauma ICU; further, we shared these modifications with our team members and educated them about these. These details have been included as a “Supplementary Material” to aid readers' comprehension. Additionally, pertinent information regarding the protocol adjustments has been incorporated into the manuscript.
- The patient groups in Table 1 should be divided into subgroups for better comparison. It is too simplified by just grouping patients by their injury mechanism as blunt/penetrating or just compare the two group of patients with their ISS score. For example, patients with gastrointestinal injury or head injury may have same ISS, but the timing of feeding can be totally different. The need of sedation and analgesics will also be different between those who underwent surgery or conservative treatment even though both group of patients may have a similar ISS.
Response: We acknowledge the reviewer's suggestion for subgroup analysis. However, as mentioned, there were no standardized criteria for subgrouping based on factors such as gastrointestinal or head injury, and surgical versus conservative treatment. Moreover, the limited number of cases within each subgroup impeded comprehensive comparison. We have included information regarding these constraints in the Discussion section.
- Therefore, patient groups from before-FD and after-FD should be well-matched before undergoing any statistical analysis.
Response: We agree with the reviewer's observation. Given the retrospective nature of our study conducted in a single institution, achieving well-matched groups for before-and-after comparisons was challenging. Additionally, there were the aforementioned challenges in the subgrouping of patients. Consequently, to mitigate these limitations, we employed multivariable logistic regression analysis, incorporating factors such as ISS and initial physiologic parameters, which we deemed most reflective of the characteristics of patients with severe trauma. We appreciate the reviewer's understanding of these constraints, which have been elaborated in the Discussion section.

Reviewer 2 Report
Comments and Suggestions for Authors
Main points
1) In introduction, you wrote “In this study, we hypothesized that the implementation of the FAST HUGS BID checklist in the TICU could reduce in-hospital mortality and length of stay (LOS) in the 56 ICU and hospital.” What is the rationale behind your hypothesis? Hypothesis development should be based on prior literature.
o What you wrote in most of the discussion should have been in the hypothesis development section, not in the discussion.
o That being said, the discussion should focus more on what your findings indicate differs or is similar to previous findings, and why you believe that is the case.
o Please rewrite the introduction and discussion, and include hypothesis development if necessary.
2) What are your theoretical contributions?
o If I understand correctly, it appears that while no studies have examined the effect of FAST HUGS BID in its entirety, previous studies have individually examined its components. Besides consolidating these previous findings, what truly constitutes novelty in this study? Were there any unexpected discoveries? If so please highlight them.
o If your focus is more on the all 11 items’, is there a rationale behind the necessity for all 11 checklist items to be met, rather than just a few?
3) Study design: What you have done is compare the pre- and post-implementation phases of the FAST HUGS BID. However, implementation does not guarantee that all checklist items are met. Essentially, your data does not indicate whether the FAST HUGS BID was successfully implemented. With that said, how can you claim that your study truly examined the effect of FAST HUGS BID? This could be one of your limitations
4) Pre-implementation: Were there no structured protocols in place before this implementation? In other words, did healthcare providers not adhere to any of the components of FAST HUGS BID? If there were existing protocols guiding providers to address certain aspects of FAST HUGS BID, what exactly does the comparison capture?
Minor points
1) In your study, FAST HUGS BID is abbreviated as FD. Please specify this abbreviation in the abstract. For example: 'FAST HUGS BID (Feeding, Analgesia, Sedation, Thromboembolic prophylaxis, Head-of-bed elevation to 10 degrees, Ulcer prophylaxis, Glycemic control, Spontaneous breathing trial, Bowel regimen, Indwelling catheter removal, and De-escalation of antibiotics) — abbreviated as FD hereafter
2) Please use either TICU or ICU in your manuscript.
3) Wrong page number
Comments on the Quality of English Language
Minor editing of English language required
Author Response
Response to Reviewer 2
Response: We sincerely appreciate the reviewers’ constructive comments and suggestions. After careful consideration of the comments, we have endeavored to revise the manuscript accordingly. The point-by-point responses are provided below. The revised sections of the manuscript are highlighted in yellow.
Main points
1) In introduction, you wrote “In this study, we hypothesized that the implementation of the FAST HUGS BID checklist in the TICU could reduce in-hospital mortality and length of stay (LOS) in the ICU and hospital.” What is the rationale behind your hypothesis? Hypothesis development should be based on prior literature.
o What you wrote in most of the discussion should have been in the hypothesis development section, not in the discussion.
o That being said, the discussion should focus more on what your findings indicate differs or is similar to previous findings, and why you believe that is the case.
o Please rewrite the introduction and discussion, and include hypothesis development if necessary.
Response: We have restructured both the Introduction and Discussion sections. In the Introduction section, we have added an additional paragraph to delineate the rationale behind our hypothesis, incorporating relevant literature references. Moreover, in the Discussion section, we have focused more on delineating the similarities and differences between our findings and previous research, highlighting the significance of our study's exclusive focus on patients with severe trauma and the comprehensive analysis of all 11 components of the FAST HUGS BID checklist.
2) What are your theoretical contributions?
- If I understand correctly, it appears that while no studies have examined the effect of FAST HUGS BID in its entirety, previous studies have individually examined its components. Besides consolidating these previous findings, what truly constitutes novelty in this study? Were there any unexpected discoveries? If so please highlight them.
- If your focus is more on the all 11 items’, is there a rationale behind the necessity for all 11 checklist items to be met, rather than just a few?
Response: Notably, we exclusively focused on critically injured patients with trauma, while previous studies included broader ICU populations. Additionally, although previous studies examined individual components of the FAST HUGS BID checklist, we performed a comprehensive analysis of all 11 checklist items. As mentioned in the Discussion section, when Vincent initially proposed the ICU checklist, it consisted of the seven-item "FAST HUG." Subsequently, it was considered more effective to augment it with four additional "S BID" items, which yielded the comprehensive "FAST HUGS BID" checklist with 11 components. Our findings support the application of the ICU bundle approach, where incorporating all 11 items of the FAST HUGS BID checklist did not significantly increase the time required for verification compared with that with the seven-item checklist. We believed that utilizing all 11 items would yield greater benefits in patient care outcomes. These points have been emphasized in the Discussion section to underscore the novelty of our study.
3) Study design: What you have done is compare the pre- and post-implementation phases of the FAST HUGS BID. However, implementation does not guarantee that all checklist items are met. Essentially, your data does not indicate whether the FAST HUGS BID was successfully implemented. With that said, how can you claim that your study truly examined the effect of FAST HUGS BID? This could be one of your limitations.
Response: We acknowledge the validity of reviewer 2's concern regarding the successful implementation of the FAST HUGS BID checklist. Although it is challenging to ensure complete adherence to all checklist items, we meticulously trained our team members on the protocol and implemented monitoring mechanisms to ensure compliance. This limitation has been duly noted in the discussion section, along with a detailed explanation of the monitoring and assessment processes post-implementation.
4) Pre-implementation: Were there no structured protocols in place before this implementation? In other words, did healthcare providers not adhere to any of the components of FAST HUGS BID? If there were existing protocols guiding providers to address certain aspects of FAST HUGS BID, what exactly does the comparison capture?
Response: Prior to implementation, there were no comprehensive protocols in place for addressing the various components of the FAST HUGS BID checklist. Healthcare providers often addressed specific aspects based on individual preferences and capabilities. Moreover, the absence of a dedicated trauma ICU team during the pre-implementation phase further compounded the challenges. We have elaborated on these limitations in the discussion section to provide a comprehensive understanding of our study's context.
Minor points
1) In your study, FAST HUGS BID is abbreviated as FD. Please specify this abbreviation in the abstract. For example: 'FAST HUGS BID (Feeding, Analgesia, Sedation, Thromboembolic prophylaxis, Head-of-bed elevation to 10 degrees, Ulcer prophylaxis, Glycemic control, Spontaneous breathing trial, Bowel regimen, Indwelling catheter removal, and De-escalation of antibiotics) — abbreviated as FD hereafter.
Response: We have revised the abstract accordingly to specify the abbreviation for FAST HUGS BID as FD.
2) Please use either TICU or ICU in your manuscript.
Response: We have standardized the term as ICU throughout the manuscript, replacing TICU with trauma ICU where appropriate.
3) Wrong page number
Response: We have rectified the mentioned discrepancy. We apologize for any confusion caused.
Comments on the Quality of English Language
Minor editing of English language required.
Response: Following the revisions based on the reviewers' comments, the manuscript underwent professional English language editing to ensure clarity and coherence.

Reviewer 3 Report
Comments and Suggestions for Authors
Study Summary:
The manuscript Impact of a Rounding Checklist Implementation in the Trauma 2 Intensive Care Unit on Clinical Outcomes, by Seo and colleagues Is a before- and after-study evaluating the impact of implementation of a rounding checklist (FASTHUGBID) on outcomes in a trauma ICU.
The authors evaluated outcomes in 696 patients pre-intervention and 1720 patients post-intervention. There was a one year gap between the two study periods during which the rounding checklist was introduced and revisions were made to its utilization. The authors state that the checklist was posted at patients' bedsides though exactly how the checklist was utilized is not described. The checklist is intended to prompt providers to discuss specific aspect of care during rounds (Feeding, Analgesia, Sedation, Thromboembolic prophylaxis, Head of bed elevation, Ulcer prophylaxis, Glycemic Control, Spontaneous Breathing Trial, Bowel Regimen, Indwelling Catheter Removal, and De-escalation of Antibiotics). The authors hypothesized that implementation of the checklist would reduce in-hospital mortality and length of stay in the ICU and hospital. The authors also evaluated certain practice characteristics associated with the checklist before and after its implementation (for example, time to initiation of feeds).
Patient characteristics compared in both groups demonstrate similar age, sex, and mechanisms of injury. Patients in the post-intervention group had a higher percentage of underlying disease and higher injury severity score. There was a statistically significant lower in-hospital mortality, percentage of patients with complications, ICU length of stay, hospital length of stay, and duration of mechanical ventilation in patients cared for after introduction of the checklist.
Ratings by characteristic:
· Novelty: This is the first large evaluation of the impact of the expanded checklist (FASTHUGBID) in the TICU setting.
· Scope: The study does fit within the scope of the journal
· Significance: Determining the impact of the checklist is very important and could have a large impact of management of patients worldwide.
· Quality: The article is generally well written and tables are clear. Some specific feedback on content is listed below.
· Scientific Soundsness: Concerns about methods are described below.
· Applicability: The limited description of how the checklist was implemented limits its applicability somewhat.
· Interest to readers: high interest
· Overall Merit: worth consideration for publication with major revision
· English Level: good
Limitations on Study Conclusions:
The study's conclusions are limited by the before- and after- study design as there is potential that evolving practice patterns independent of the checklist impacted outcomes. The authors do describe practice characteristics related to each of the checklist's components in the pre- and post-intervention periods. It should be noted that best practices related to each of the checklist's components are not specified in the study description. However, the authors do describe existing data supporting best practices in the discussion section. There are a number of statistically significant differences in practices (for example, shorter time to initiation of enoxaprin) before and after the intervention. While the use of the checklist may have had an impact on this practice pattern, the relationship is limited to an association.
Grammar/Structure of Discussion:
There are several minor grammatical concerns:
Abstract: line 21 has "." after ventilation; grammar and sentence structure here should be corrected.
Introduction: The authors should consider consolidating their introduction - particularly the first paragraph. There are several awkwardly worded sentences. For example, lines 31-32: "Such errors can arise from various factors, including unstable vital signs, incomplete histories, and insufficient information." This sentence is vague. Lines 38-29: "Moreover, ensuring a seamless continuity of management within the same quality of surgical services contributes to improved outcomes." Consider structuring for clarity.
Additional feedback by section:
Methods section:
It is unclear why the number of patients screened included patients admitted to the wards. Authors should describe how "immediate non-survivor" was defined. In Figure 1, the legend should define "DOA."
There is a concerning gap in time between the pre-intervention period and the post-intervention period. These periods are defined as: pre-intervention period: March 2016-Dec 2017 and post-intervention period: Jan 2019-Dec 2020. The checklist was introduced in "mid" 2018. Patient outcomes in 2018 were not included in the study. It is not clear why the authors did not include patients in the pre-intervention analysis up to the date at which the checklist was introduced. From a quality improvement standpoint, it would have been valuable to have more information about the impact on outcomes immediately following introduction of the checklist, including changes in implementation strategies. This type of information would benefit other programs planning to use this or similar checklists.
Along the same lines, it would be valuable to know what methods were used to educate providers about the checklist and how it was practically utilized on rounds. It would also be valuable to know how prescriptive individual components of the checklist were, if at all. For example, were there guidelines for providers regarding nutritional practices.
Results:
There is an impressive difference in the Primary Outcomes assessed before and after the introduction of the checklist. These findings would be even stronger if data during the year 2018 was included in the analysis. It is intriguing that the incidence of sepsis was higher in the post-intervention period. The authors do not speculate on this observation. The comparison of individual components of the checklist is also presented. While there are observable differences in several important aspects of management, such as analgesia and sedation, it is difficult to know if these differences are attributable to the checklist versus changes in practice patterns influenced by other factors.
Why are the number so different. Please add in the number of total admits in each time periods and the number with percent of exclusions. Please add in figure 1 the numbers of patients excluded for each
Reason in each time period. For example age <18yo (4 period 1, 8 period 2 or whatever. Although the time periods were 22 vs 24 month you have more than double the sample size. Why is this. Add this as a limitation.
Discussion Section:
The authors discuss their primary findings then proceed to discuss each individual component of the Checklist. Two recommendations regarding this section. 1) The authors should consider reflecting on the published data they include and consolidate the information. There are a number of sections which include several sentences about standards of care which could be shortened. I would also encourage the authors to be conservative about their inferences. For example, on page 4/16, line 272, can it be inferred that higher utilization of LMWH and enoxaparin and shorter time to administration of enoxaparin contributed to improved clinical outcomes? For this inference specifically, the authors report that there was no difference in venous thromboembolism in Table 3.
The authors do note several important limitations, including the gap in patient data in 2018.
In conclusion, I think that this manuscript warrants reconsideration for publication with major revisions, including:
-description of rational behind lack of data collection in 2018
-additional description of implementation strategies and lessons learned in the introduction
-re-assessment of inferences of impact of individual components on outcomes of each aspect of the Checklist in discussion section
Comments on the Quality of English LanguageStudy Summary:
The manuscript Impact of a Rounding Checklist Implementation in the Trauma 2 Intensive Care Unit on Clinical Outcomes, by Seo and colleagues Is a before- and after-study evaluating the impact of implementation of a rounding checklist (FASTHUGBID) on outcomes in a trauma ICU.
The authors evaluated outcomes in 696 patients pre-intervention and 1720 patients post-intervention. There was a one year gap between the two study periods during which the rounding checklist was introduced and revisions were made to its utilization. The authors state that the checklist was posted at patients' bedsides though exactly how the checklist was utilized is not described. The checklist is intended to prompt providers to discuss specific aspect of care during rounds (Feeding, Analgesia, Sedation, Thromboembolic prophylaxis, Head of bed elevation, Ulcer prophylaxis, Glycemic Control, Spontaneous Breathing Trial, Bowel Regimen, Indwelling Catheter Removal, and De-escalation of Antibiotics). The authors hypothesized that implementation of the checklist would reduce in-hospital mortality and length of stay in the ICU and hospital. The authors also evaluated certain practice characteristics associated with the checklist before and after its implementation (for example, time to initiation of feeds).
Patient characteristics compared in both groups demonstrate similar age, sex, and mechanisms of injury. Patients in the post-intervention group had a higher percentage of underlying disease and higher injury severity score. There was a statistically significant lower in-hospital mortality, percentage of patients with complications, ICU length of stay, hospital length of stay, and duration of mechanical ventilation in patients cared for after introduction of the checklist.
Ratings by characteristic:
· Novelty: This is the first large evaluation of the impact of the expanded checklist (FASTHUGBID) in the TICU setting.
· Scope: The study does fit within the scope of the journal
· Significance: Determining the impact of the checklist is very important and could have a large impact of management of patients worldwide.
· Quality: The article is generally well written and tables are clear. Some specific feedback on content is listed below.
· Scientific Soundsness: Concerns about methods are described below.
· Applicability: The limited description of how the checklist was implemented limits its applicability somewhat.
· Interest to readers: high interest
· Overall Merit: worth consideration for publication with major revision
· English Level: good
Limitations on Study Conclusions:
The study's conclusions are limited by the before- and after- study design as there is potential that evolving practice patterns independent of the checklist impacted outcomes. The authors do describe practice characteristics related to each of the checklist's components in the pre- and post-intervention periods. It should be noted that best practices related to each of the checklist's components are not specified in the study description. However, the authors do describe existing data supporting best practices in the discussion section. There are a number of statistically significant differences in practices (for example, shorter time to initiation of enoxaprin) before and after the intervention. While the use of the checklist may have had an impact on this practice pattern, the relationship is limited to an association.
Grammar/Structure of Discussion:
There are several minor grammatical concerns:
Abstract: line 21 has "." after ventilation; grammar and sentence structure here should be corrected.
Introduction: The authors should consider consolidating their introduction - particularly the first paragraph. There are several awkwardly worded sentences. For example, lines 31-32: "Such errors can arise from various factors, including unstable vital signs, incomplete histories, and insufficient information." This sentence is vague. Lines 38-29: "Moreover, ensuring a seamless continuity of management within the same quality of surgical services contributes to improved outcomes." Consider structuring for clarity.
Additional feedback by section:
Methods section:
It is unclear why the number of patients screened included patients admitted to the wards. Authors should describe how "immediate non-survivor" was defined. In Figure 1, the legend should define "DOA."
There is a concerning gap in time between the pre-intervention period and the post-intervention period. These periods are defined as: pre-intervention period: March 2016-Dec 2017 and post-intervention period: Jan 2019-Dec 2020. The checklist was introduced in "mid" 2018. Patient outcomes in 2018 were not included in the study. It is not clear why the authors did not include patients in the pre-intervention analysis up to the date at which the checklist was introduced. From a quality improvement standpoint, it would have been valuable to have more information about the impact on outcomes immediately following introduction of the checklist, including changes in implementation strategies. This type of information would benefit other programs planning to use this or similar checklists.
Along the same lines, it would be valuable to know what methods were used to educate providers about the checklist and how it was practically utilized on rounds. It would also be valuable to know how prescriptive individual components of the checklist were, if at all. For example, were there guidelines for providers regarding nutritional practices.
Results:
There is an impressive difference in the Primary Outcomes assessed before and after the introduction of the checklist. These findings would be even stronger if data during the year 2018 was included in the analysis. It is intriguing that the incidence of sepsis was higher in the post-intervention period. The authors do not speculate on this observation. The comparison of individual components of the checklist is also presented. While there are observable differences in several important aspects of management, such as analgesia and sedation, it is difficult to know if these differences are attributable to the checklist versus changes in practice patterns influenced by other factors.
Why are the number so different. Please add in the number of total admits in each time periods and the number with percent of exclusions. Please add in figure 1 the numbers of patients excluded for each
Reason in each time period. For example age <18yo (4 period 1, 8 period 2 or whatever. Although the time periods were 22 vs 24 month you have more than double the sample size. Why is this. Add this as a limitation.
Discussion Section:
The authors discuss their primary findings then proceed to discuss each individual component of the Checklist. Two recommendations regarding this section. 1) The authors should consider reflecting on the published data they include and consolidate the information. There are a number of sections which include several sentences about standards of care which could be shortened. I would also encourage the authors to be conservative about their inferences. For example, on page 4/16, line 272, can it be inferred that higher utilization of LMWH and enoxaparin and shorter time to administration of enoxaparin contributed to improved clinical outcomes? For this inference specifically, the authors report that there was no difference in venous thromboembolism in Table 3.
The authors do note several important limitations, including the gap in patient data in 2018.
In conclusion, I think that this manuscript warrants reconsideration for publication with major revisions, including:
-description of rational behind lack of data collection in 2018
-additional description of implementation strategies and lessons learned in the introduction
-re-assessment of inferences of impact of individual components on outcomes of each aspect of the Checklist in discussion section
Author Response
Response to Reviewer 3
Response: We appreciate the comprehensive evaluation of our manuscript and the valuable feedback provided by reviewer 3. In response to the concerns and suggestions raised, we have thoroughly revised the manuscript to address each point. The specific revisions and responses to the reviewer's comments are outlined below.
Study Summary:
The manuscript Impact of a Rounding Checklist Implementation in the Trauma Intensive Care Unit on Clinical Outcomes, by Seo and colleagues Is a before- and after-study evaluating the impact of implementation of a rounding checklist (FASTHUGBID) on outcomes in a trauma ICU.
The authors evaluated outcomes in 696 patients pre-intervention and 1720 patients post-intervention. There was a one year gap between the two study periods during which the rounding checklist was introduced and revisions were made to its utilization. The authors state that the checklist was posted at patients' bedsides though exactly how the checklist was utilized is not described. The checklist is intended to prompt providers to discuss specific aspect of care during rounds (Feeding, Analgesia, Sedation, Thromboembolic prophylaxis, Head of bed elevation, Ulcer prophylaxis, Glycemic Control, Spontaneous Breathing Trial, Bowel Regimen, Indwelling Catheter Removal, and De-escalation of Antibiotics). The authors hypothesized that implementation of the checklist would reduce in-hospital mortality and length of stay in the ICU and hospital. The authors also evaluated certain practice characteristics associated with the checklist before and after its implementation (for example, time to initiation of feeds).
Patient characteristics compared in both groups demonstrate similar age, sex, and mechanisms of injury. Patients in the post-intervention group had a higher percentage of underlying disease and higher injury severity score. There was a statistically significant lower in-hospital mortality, percentage of patients with complications, ICU length of stay, hospital length of stay, and duration of mechanical ventilation in patients cared for after introduction of the checklist.
Ratings by characteristic:
- Novelty: This is the first large evaluation of the impact of the expanded checklist (FASTHUGBID) in the TICU setting.
- Scope: The study does fit within the scope of the journal
- Significance: Determining the impact of the checklist is very important and could have a large impact of management of patients worldwide.
- Quality: The article is generally well written and tables are clear. Some specific feedback on content is listed below.
- Scientific Soundsness: Concerns about methods are described below.
- Applicability: The limited description of how the checklist was implemented limits its applicability somewhat.
- Interest to readers: high interest
- Overall Merit: worth consideration for publication with major revision
- English Level: good
Limitations on Study Conclusions:
The study's conclusions are limited by the before- and after- study design as there is potential that evolving practice patterns independent of the checklist impacted outcomes. The authors do describe practice characteristics related to each of the checklist's components in the pre- and post-intervention periods. It should be noted that best practices related to each of the checklist's components are not specified in the study description. However, the authors do describe existing data supporting best practices in the discussion section. There are a number of statistically significant differences in practices (for example, shorter time to initiation of enoxaprin) before and after the intervention. While the use of the checklist may have had an impact on this practice pattern, the relationship is limited to an association.
Grammar/Structure of Discussion:
There are several minor grammatical concerns:
Abstract: line 21 has "." after ventilation; grammar and sentence structure here should be corrected.
Introduction: The authors should consider consolidating their introduction - particularly the first paragraph. There are several awkwardly worded sentences. For example, lines 31-32: "Such errors can arise from various factors, including unstable vital signs, incomplete histories, and insufficient information." This sentence is vague. Lines 38-29: "Moreover, ensuring a seamless continuity of management within the same quality of surgical services contributes to improved outcomes." Consider structuring for clarity.
Response: As per your comment, we have further clarified the sentences.
Additional feedback by section:
Methods section:
It is unclear why the number of patients screened included patients admitted to the wards. Authors should describe how "immediate non-survivor" was defined. In Figure 1, the legend should define "DOA."
Response: Thank you for your comment. We have provided clarification regarding the definition of "immediate non-survivor" as "dead on arrival" to avoid confusion. In Figure 1, we have revised “DOA” to “Dead on arrival.”
There is a concerning gap in time between the pre-intervention period and the post-intervention period. These periods are defined as: pre-intervention period: March 2016-Dec 2017 and post-intervention period: Jan 2019-Dec 2020. The checklist was introduced in "mid" 2018. Patient outcomes in 2018 were not included in the study. It is not clear why the authors did not include patients in the pre-intervention analysis up to the date at which the checklist was introduced. From a quality improvement standpoint, it would have been valuable to have more information about the impact on outcomes immediately following introduction of the checklist, including changes in implementation strategies. This type of information would benefit other programs planning to use this or similar checklists.
Response: We acknowledge the concern regarding the gap in patient data collection in 2018. The rationale behind this gap has been clarified in the manuscript. Specifically, partial implementation of the checklist began in early 2018, with full implementation being established by 2019. This interim phase was considered as part of the checklist's implementation process; accordingly, we excluded data from 2018 to prevent bias in the before-after comparison, since the FD checklist was being refined through trial and error during that period. This limitation has been duly noted in the Discussion section.
Along the same lines, it would be valuable to know what methods were used to educate providers about the checklist and how it was practically utilized on rounds. It would also be valuable to know how prescriptive individual components of the checklist were, if at all. For example, were there guidelines for providers regarding nutritional practices.
Response: Thank you for this comment. Indeed, we modified the FAST HUGS BID protocol to align with the specific requirements of our trauma ICU; further, we shared and educated our team members on these modifications. These details have been included as a “Supplementary Material” to aid readers' comprehension. Additionally, pertinent information regarding the protocol adjustments has been incorporated into the manuscript. Although it is difficult to ensure complete adherence to all checklist items, we meticulously trained our team members on the protocol and implemented monitoring mechanisms to ensure compliance. However, we concede that our data may not fully capture the extent of checklist adherence. This limitation has been duly noted in the discussion section, along with a detailed explanation of the monitoring and assessment processes post-implementation.
Results:
There is an impressive difference in the Primary Outcomes assessed before and after the introduction of the checklist. These findings would be even stronger if data during the year 2018 was included in the analysis. It is intriguing that the incidence of sepsis was higher in the post-intervention period. The authors do not speculate on this observation. The comparison of individual components of the checklist is also presented. While there are observable differences in several important aspects of management, such as analgesia and sedation, it is difficult to know if these differences are attributable to the checklist versus changes in practice patterns influenced by other factors.
Response: Thank you for your comment. Indeed, the higher incidence of sepsis in the after-FD group may be attributed to the higher severity of trauma (ISS 13 vs. 17) and greater prevalence of comorbidities (38.3% vs. 42.9%). However, the exact reasons underlying this finding remain unclear. Additionally, although there was a statistically significant increase in the absolute number of sepsis cases, it was not deemed clinically significant. Nonetheless, to avoid confusion, we have mentioned this aspect in the manuscript. As the reviewer noted, it cannot be completely ruled out that various other factors influenced our findings. These aspects are discussed in the Discussion section as major limitations inherent in the retrospective pre-post comparison design adopted in our study.
Why are the number so different. Please add in the number of total admits in each time periods and the number with percent of exclusions. Please add in figure 1 the numbers of patients excluded for each Reason in each time period. For example age <18yo (4 period 1, 8 period 2 or whatever. Although the time periods were 22 vs 24 month you have more than double the sample size. Why is this. Add this as a limitation.
Response: Thank you for your comment. We have revised Figure 1 for clarity in accordance with your comments. Furthermore, we have addressed the concerns regarding the gap in time between the before-FD and after-FD periods. The increased sample size in the after-FD period could be attributed to the growing maturity of the trauma center, resulting in an influx of severely injured patients with trauma. This limitation has been explicitly mentioned in the Discussion section.
Discussion Section:
The authors discuss their primary findings then proceed to discuss each individual component of the Checklist. Two recommendations regarding this section. 1) The authors should consider reflecting on the published data they include and consolidate the information. There are a number of sections which include several sentences about standards of care which could be shortened. I would also encourage the authors to be conservative about their inferences. For example, on page 4/16, line 272, can it be inferred that higher utilization of LMWH and enoxaparin and shorter time to administration of enoxaparin contributed to improved clinical outcomes? For this inference specifically, the authors report that there was no difference in venous thromboembolism in Table 3.
The authors do note several important limitations, including the gap in patient data in 2018.
Response: We have revised the Discussion section to consolidate information and ensure conservative inferences. Specifically, we have removed the inference regarding the impact of LMWH and enoxaparin utilization on improved clinical outcomes. Moreover, we have included additional descriptions of implementation strategies and lessons learned in the introduction, as suggested by the reviewer.
In conclusion, I think that this manuscript warrants reconsideration for publication with major revisions, including:
-description of rational behind lack of data collection in 2018
-additional description of implementation strategies and lessons learned in the introduction
-re-assessment of inferences of impact of individual components on outcomes of each aspect of the Checklist in discussion section
Response: We sincerely appreciate the reviewers’ constructive comments and suggestions. We have addressed these aspects in our revised manuscript, with changes highlighted in yellow.

Round 2
Reviewer 2 Report
Comments and Suggestions for Authors
It would have been great if you highlighted the page and line numbers of revisions in your response to the reviewer's comments. This would ensure that you have indeed addressed the comments. Other than that, there are no further comments. Thank you for your hard work.
Comments on the Quality of English LanguageMinor editing of English language required.
Author Response
Response: Thank you for the comments and suggestions. I apologize for not highlighting the revised page and line numbers in the responses to reviewer comments during the process of incorporating revisions based on your feedback as well as comments and suggestions from two additional reviewers.
Reviewer 3 Report
Comments and Suggestions for Authors
-Figure 1:
- change to "deaths on arrival" to “dead on arrival”
-FASTHUGBID spelled out completely in some boxes and “FD” used in others, I would strive to make the abbreviations consistent
-another example of using FASTHUGBID after introducing the abbreviation FD is on page 2, line 80
Introduction:
Page 2, line 54 - consider changing "regarding" to "of"
Page 2, line 63 –“ …demonstrated that its implementation decreased the rate of catheter-related bloodstream infections through checklist utilization [11].”
-consider deleting “through checklist utilization” as this is somewhat redundant as you have already said “its implementation”
Page 2, lines 61-64: grammar concern, sentence structure does not flow: consider whether the intra-operative checklist comment and associated references are needed
“Additionally, some studies have demonstrated a reduction in the rate of ventilator-associated pneumonia (VAP) following the use of a checklist [12, 13], while Haynes et al. reported a decrease in mortality and complications following intraoperative utilization of a checklist [14, 15].
Page 2, lines 66-67:
“Accordingly, we hypothesized that the implementation of the FD checklist in the trauma ICU could reduce in-hospital mortality and length of stay (LOS) in the ICU and hospital.
-consider changing "could" be "would"
Page 5 - secondary outcomes
-consider moving the discussion/speculation about incidence of sepsis to the discussion section instead of the results section
Page 6 - line 163 - consider “oral analgesics” rather the informal term "pain-killers"
***the page numbers restart in midst of manuscript***
Discussion:
-lines 215-221: do you have data to support the highlighted conclusion?
“There are several factors demonstrating that the implementation of the FD checklist substantially affects the trauma ICU population. First, the implementation of the checklist resulted in a decrease in medical errors due to its concise and easily memorable format, which included the fundamental components of care provided by all members of the trauma team. Second, the trauma team could make daily care plans and assess their completion using the checklist during daily ICU rounds.
Line 280: grammar concern – would eliminate or reword the following sentence
“Since the 1990s, head-of-bed elevation reduces the incidence of gastroesophageal reflux and VAP in ventilated patients [28].”
Line 287: (30°–60°) -> insert space after the dash
Lines: 319-329: “S for Spontaneous Breathing Trial. …In this study, we observed that the after-FD group had a shorter time to extubation but a higher rate of unplanned intubation than the before-FD group. Herein, we need to interpret with caution that the before-FD group exhibited an unusually low rate (0.9%) of unplanned intubation, which is exceptionally lower than the optimal range for failed extubation (5%–10%) [37]. These findings paradoxically suggest that the absence of daily spontaneous awakening or breathing trials before the implementation of the checklist might have contributed to prolonged ventilator care.”
-this wording in this paragraph regarding spontaneous breathing trials is a bit confusing. I would suggest limiting your interpretations to something along the lines that the decreased incidence of re-intubation pre-FD may have been related to more cautious practices regarding extubation attempts.
Line 348
“Consequently, these efforts might have reduced the LOS in the ICU and hospital as well as improved mortality.”
Consider changing “improved” to “reduced”
Comments on the Quality of English Language
-Figure 1:
- change to "deaths on arrival" to “dead on arrival”
-FASTHUGBID spelled out completely in some boxes and “FD” used in others, I would strive to make the abbreviations consistent
-another example of using FASTHUGBID after introducing the abbreviation FD is on page 2, line 80
Introduction:
Page 2, line 54 - consider changing "regarding" to "of"
Page 2, line 63 –“ …demonstrated that its implementation decreased the rate of catheter-related bloodstream infections through checklist utilization [11].”
-consider deleting “through checklist utilization” as this is somewhat redundant as you have already said “its implementation”
Page 2, lines 61-64: grammar concern, sentence structure does not flow: consider whether the intra-operative checklist comment and associated references are needed
“Additionally, some studies have demonstrated a reduction in the rate of ventilator-associated pneumonia (VAP) following the use of a checklist [12, 13], while Haynes et al. reported a decrease in mortality and complications following intraoperative utilization of a checklist [14, 15].
Page 2, lines 66-67:
“Accordingly, we hypothesized that the implementation of the FD checklist in the trauma ICU could reduce in-hospital mortality and length of stay (LOS) in the ICU and hospital.
-consider changing "could" be "would"
Page 5 - secondary outcomes
-consider moving the discussion/speculation about incidence of sepsis to the discussion section instead of the results section
Page 6 - line 163 - consider “oral analgesics” rather the informal term "pain-killers"
***the page numbers restart in midst of manuscript***
Discussion:
-lines 215-221: do you have data to support the highlighted conclusion?
“There are several factors demonstrating that the implementation of the FD checklist substantially affects the trauma ICU population. First, the implementation of the checklist resulted in a decrease in medical errors due to its concise and easily memorable format, which included the fundamental components of care provided by all members of the trauma team. Second, the trauma team could make daily care plans and assess their completion using the checklist during daily ICU rounds.
Line 280: grammar concern – would eliminate or reword the following sentence
“Since the 1990s, head-of-bed elevation reduces the incidence of gastroesophageal reflux and VAP in ventilated patients [28].”
Line 287: (30°–60°) -> insert space after the dash
Lines: 319-329: “S for Spontaneous Breathing Trial. …In this study, we observed that the after-FD group had a shorter time to extubation but a higher rate of unplanned intubation than the before-FD group. Herein, we need to interpret with caution that the before-FD group exhibited an unusually low rate (0.9%) of unplanned intubation, which is exceptionally lower than the optimal range for failed extubation (5%–10%) [37]. These findings paradoxically suggest that the absence of daily spontaneous awakening or breathing trials before the implementation of the checklist might have contributed to prolonged ventilator care.”
-this wording in this paragraph regarding spontaneous breathing trials is a bit confusing. I would suggest limiting your interpretations to something along the lines that the decreased incidence of re-intubation pre-FD may have been related to more cautious practices regarding extubation attempts.
Line 348
“Consequently, these efforts might have reduced the LOS in the ICU and hospital as well as improved mortality.”
Consider changing “improved” to “reduced”
Author Response
Response to Reviewer 3
Response: We appreciate the comprehensive evaluation of our manuscript and the valuable feedback provided by Reviewer 3. In response to the concerns and suggestions raised, we have thoroughly revised the manuscript to address each point. The specific revisions and responses to the reviewer's comments are outlined below.
Comments and Suggestions for Authors
-Figure 1:
- change to "deaths on arrival" to “dead on arrival”
-FASTHUGBID spelled out completely in some boxes and “FD” used in others, I would strive to make the abbreviations consistent
-another example of using FASTHUGBID after introducing the abbreviation FD is on page 2, line 80
Response: We have revised the Figure 1 and the text body according to your comments.
Introduction:
Page 2, line 54 - consider changing "regarding" to "of"
Response: We have revised it accordingly.
Page 2, line 63 –“ …demonstrated that its implementation decreased the rate of catheter-related bloodstream infections through checklist utilization [11].”
-consider deleting “through checklist utilization” as this is somewhat redundant as you have already said “its implementation”
Response: We have deleted “through checklist utilization” in accordance with your comments.
Page 2, lines 61-64: grammar concern, sentence structure does not flow: consider whether the intra-operative checklist comment and associated references are needed
“Additionally, some studies have demonstrated a reduction in the rate of ventilator-associated pneumonia (VAP) following the use of a checklist [12, 13], while Haynes et al. reported a decrease in mortality and complications following intraoperative utilization of a checklist [14, 15].
Response: According your comment, we have modified this sentence for better flow and clarity as follows.
“Additionally, some studies have demonstrated a reduction in the rate of ventilator-associated pneumonia (VAP) following the use of a checklist [12, 13]. Haynes et al. also reported a decrease in mortality and complications following intraoperative utilization of a checklist [14, 15].”
Page 2, lines 66-67:
“Accordingly, we hypothesized that the implementation of the FD checklist in the trauma ICU could reduce in-hospital mortality and length of stay (LOS) in the ICU and hospital.
-consider changing "could" be "would"
Response: We have revised it accordingly.
Page 5 - secondary outcomes
-consider moving the discussion/speculation about incidence of sepsis to the discussion section instead of the results section
Response: We have modified some of the discussion/speculation about incidence of sepsis and moved it from the result section to the discussion section according to your comment.
Page 6 - line 163 - consider “oral analgesics” rather the informal term "pain-killers"
Response: We have revised the term.
***the page numbers restart in midst of manuscript***
Response: Sorry for the confusion. We revised the page numbers.
Discussion:
-lines 215-221: do you have data to support the highlighted conclusion?
“There are several factors demonstrating that the implementation of the FD checklist substantially affects the trauma ICU population. First, the implementation of the checklist resulted in a decrease in medical errors due to its concise and easily memorable format, which included the fundamental components of care provided by all members of the trauma team. Second, the trauma team could make daily care plans and assess their completion using the checklist during daily ICU rounds.
Response: We have provided references to the highlighted conclusion.
Line 280: grammar concern – would eliminate or reword the following sentence
“Since the 1990s, head-of-bed elevation reduces the incidence of gastroesophageal reflux and VAP in ventilated patients [28].”
Response: We have made the necessary revisions to improve clarity and accuracy.
Line 287: (30°–60°) -> insert space after the dash
Response: We have inserted space after the dash.
Lines: 319-329: “S for Spontaneous Breathing Trial. …In this study, we observed that the after-FD group had a shorter time to extubation but a higher rate of unplanned intubation than the before-FD group. Herein, we need to interpret with caution that the before-FD group exhibited an unusually low rate (0.9%) of unplanned intubation, which is exceptionally lower than the optimal range for failed extubation (5%–10%) [37]. These findings paradoxically suggest that the absence of daily spontaneous awakening or breathing trials before the implementation of the checklist might have contributed to prolonged ventilator care.”
-this wording in this paragraph regarding spontaneous breathing trials is a bit confusing. I would suggest limiting your interpretations to something along the lines that the decreased incidence of re-intubation pre-FD may have been related to more cautious practices regarding extubation attempts.
Response: We apologize for the confusion caused. Following your suggestion, we have revised the paragraph to make it clearer.
Line 348
“Consequently, these efforts might have reduced the LOS in the ICU and hospital as well as improved mortality.”
Consider changing “improved” to “reduced”
Response: We have changed it accordingly.